# Visits to Sexually Transmitted Infection Clinics in Italy from January 2016 to November 2021: A Multicenter, Retrospective Study

**DOI:** 10.3390/jpm13050731

**Published:** 2023-04-26

**Authors:** Alessandro Borghi, Maria Elena Flacco, Lucrezia Pacetti, Gionathan Orioni, Elisa Marzola, Rosario Cultrera, Valentina Guerra, Roberto Manfredini, Valeria Gaspari, Daniela Segala, Monica Corazza

**Affiliations:** 1Department of Medical Sciences, Section of Dermatology and Infectious Diseases, University of Ferrara, 44121 Ferrara, Italy; 2Department of Environmental and Prevention Sciences, University of Ferrara, 44121 Ferrara, Italy; 3Dermatology Unit, IRCCS Azienda Ospedaliero-Universitaria di Bologna, 40126 Bologna, Italy; 4Department of Medical and Surgical Sciences, Alma Mater Studiorum University of Bologna, 40126 Bologna, Italy; 5Infectious Diseases, Department of Translational Medicine, University of Ferrara, 44121 Ferrara, Italy; 6Planning and Management Control, University Hospital of Ferrara, 44124 Cona, Italy; 7University Center for Studies on Gender Medicine, Department of Medical Sciences, University of Ferrara, 44121 Ferrara, Italy

**Keywords:** sexually transmitted infections, seasonality, COVID-19, pandemic, gender

## Abstract

There is no evidence of seasonal variation in visits to clinics dedicated to sexually transmitted infections (STIs) in Italy, nor of changes after the advent of the COVID-19 pandemic. An observational, retrospective, multicentric study was conducted to record and analyze all the visits to the STI clinics of the Dermatology Units of the University Hospitals of Ferrara and Bologna and of the Infectious Disease Unit of Ferrara, Italy, between January 2016 and November 2021. Overall, 11.733 visits were registered over a 70-month study period (63.7% males, mean age 34.5 ± 12.8 yrs). The mean number of monthly visits significantly decreased from the advent of the pandemic (136) compared to before (177). In the pre-pandemic period, visits to STI clinics increased in the autumn/winter months when compared to spring/summer, while the trend was the opposite in the pandemic period. Thus, during the pandemic, both an overall significant reduction in visits to STI clinics and a reversal in their seasonality were observed. These trends affected males and females equally. The marked decrease, mostly found in the pandemic winter months, can be linked to the “lockdown”/self-isolation ordinances and social distancing measures during the colder months, coinciding with the spread of the COVID-19 infection, which limited the opportunities for meeting and socializing.

## 1. Introduction

Sexually transmitted infections (STIs), also known as sexually transmitted diseases, involve the transmission of an organism, either bacterial, viral, or parasites, with the exchange of bodily fluids from an infected person, or, more broadly, through intimate physical contact [1,2].

Transmission of STIs can include different routes of sexual contact, either oral, anal, vaginal or any other mucosal surfaces, mainly through microscopic abrasions of the mucosal lining [3].

The use of intravenous drugs, exposure through the vagina during childbirth, or breastfeeding can be further routes of transmission, although less frequent [3].

The most common infections are human papillomavirus, genital herpes, syphilis, human immunodeficiency virus, gonorrhea, chlamydia and trichomoniasis, with wide variations in prevalence among different countries [1,3].

The presenting condition, its signs and symptoms, depend on the specific organism and the route through which the infection spreads. Risk factors that increase the probability of transmission of STIs include unprotected sex, multiple partners, personal history of STIs, sexual assault, and use of alcohol and drugs, especially intravenous.

STIs have a high incidence in most countries, especially between 15 and 50 years of age. They can affect all types of people and can be prevented with proper education and barrier control [4,5].

STIs are a worldwide health problem and a concern and burden on healthcare systems [1].

In Italy, data are provided by the Italian National Institute of Health (INIH) and reported to the European Centre for Disease Prevention and Control (ECDC) [6,7].

So far, only a handful of studies worldwide have investigated trends in STIs diagnosis and/or changes in sexual behavior throughout the year and in relation to the seasons [8,9,10,11,12,13,14,15]. A previous study from Australia investigated which factors lead to fluctuations in numbers of daily visits to STI clinics from a seasonal/meteorological point of view, d based on the demographic. It showed that more patients accessed the services on Monday mornings and Friday afternoons than any other time of the week and that the first four months of the year have more visits, especially January. More patients tend to access STI clinics when the temperatures are milder (20–29 °C) but with more intense rainfall (>5 mL). Lastly, school vacation days and the week following a bank holiday are the busiest [16].

To date, no study has investigated trends in terms of access to STIs-dedicated public health services in Italy. If there is a seasonal difference in rates of access to STI clinics, then public health campaigns could be most effective if timed accordingly. Learning seasonal-specific trends in visits can lead to a better allocation of resources to tackle the needs of the users, especially in open-access services and clinics.

Severe acute respiratory syndrome coronavirus 2 (SARS-CoV-2) has quickly spread worldwide since the Wuhan Municipal Health Commission (China) informed the World Health Organization (WHO) of a novel pneumonia of unknown etiology in December 2019.

On 8 March 2020, an unprecedented self-isolation ordinance was launched by the Italian government, influencing numerous aspects of the social life of the Italian population [17].

Strict “lockdown” measures lasted until late April 2020, but life did not go back to normal. Softer and stricter ordinances alternated in the following months, depending on the trends/stats of new COVID-19 cases; each Italian region ruled independently on the matter. The measures were especially strict during the winter months, due to higher numbers of new cases.

Social distancing measures lasted even longer than the self-isolation and lockdown ordinances, some still being present today (2023). These measures caused deep changes in social norms, with an impact we still have to this day [18,19].

The effects of restrictions due to COVID-19 put in place on sexual habits and access to reproductive health clinics are debated. After the introduction of COVID-19 lockdowns and restrictions globally, two polar-opposite theories about sexual and reproductive health surfaced. Some experts suggested that social and travel restrictions could have caused a reduced frequency in sexual intercourse; this could prove to be a unique opportunity to reduce the burden of STIs and improve their control [20]. However, other researchers speculated that COVID-19 restrictions would have encouraged condomless sex and intimate partner violence (IPV) [21].

Some of us recently tackled this debated topic [22]. A decrease in the number of outpatient consultations, together with a substantial change in the kind of clinical questions put by the patient, was noticed during the pandemic, maybe linked to a real decrease in the number of cases of STIs. This could be due to: (1) limitation of social contact; (2) persisting stress caused by the pandemic, which may lead to a deterioration in sexual function, especially in female patients; and (3) the fact that the knowledge that COVID-19 could be transmitted through sexual intercourses can be off-putting for some individuals, reducing risky sexual behavior.

Another potential effect of the COVID-19 pandemic could be the closure of important sexual and reproductive health centers: staff and public resources had to be reallocated to tackle the pandemic, leaving “non-essential” services neglected [23]. The temporary closure of some sexual health services, due to their designation as “non-essential”, negatively impacted the patients, some of whom had unmet sexual health needs for the duration of the stricter measure of lockdown. This could have led to a delayed diagnosis, which could have had relevant repercussions on the general health and quality of life of the patient.

In line with this, a study from the Netherlands investigated the impact of the COVID-19 pandemic on sexual healthcare use, pre-exposure prophylaxis (PrEP) use and sexually-transmitted-infection incidence among men who have sex with men. This study found that both sexual health care and PrEP use decreased concurrently with the pandemic [24].

For the aforementioned reasons, the Dermatology Units for STIs of the University Hospitals of Ferrara and Bologna, Northern Italy, together with the STIs service of the Infectious Disease Unit of the University of Ferrara, joined forces on a retrospective, observational study to determine what trends, if any, there are in the matter of seasonality in terms of the number of visits to STI clinics.

We also decided to investigate the difference in terms of accesses to the STI clinics before and during the pandemic.

## 2. Materials and Methods

### 2.1. Study Design and Setting

This was an observational, retrospective, multicentric study, involving the Dermatology Units of the University Hospitals of Ferrara and Bologna, which provide a free-access service from Monday to Friday, and the Infectious Disease Unit of Ferrara. We decided to include the cases of the last 5 years of all 3 centers.

The study was based on the gathering of specific data relevant to each visit made to our STI units between January 2016 and November 2021 (included). This research was approved by the local Ethics Committee (protocol CE-AVEC 678/2022/Oss/AOUFe). A waiver of informed consent was granted for the subjects whose consent was impossible to obtain because they were untraceable. For data collection and elaboration, a system of pseudo-anonymization, in which the study subjects were assigned a code, was adopted.

### 2.2. Population and Assessments

For each visit, age (divided into age brackets <20, 20–29, 30–39, 40–49, 50–59 and >60), gender, and geographic origin (Italy vs. outside-Italy) of the patient were recorded on an Excel database. A diagnosis could not be reached for each visit, and was therefore not included in the variables in the study.

The subjects included were adults who voluntarily turned to our services dedicated to the diagnosis and care of STIs, without the need for a scheduled appointment.

If the same patient returned to our clinics more than once over the 5-year period, each new visit was registered separately as an individual account.

The exclusion criteria were the following: patients under the age of 18 and patients who turned to other dermatological clinics, even within the same units.

### 2.3. Statistical Analysis

Potential differences in the selected variables between pre-pandemic and pandemic periods were evaluated using standard univariate analyses (chi-squared test and *t*-test for categorical and continuous variables, respectively). In order to investigate the association between the number of visits due to STIs and pandemic onset, a random-effects linear regression model was fit, with the number of visits as the dependent variable, and the period (pre-pandemic vs. pandemic), age, gender, nationality and season as the independent variables. In the model, we included one cluster unit (hospital ward: Bologna or Ferrara). Statistical significance was set as a two-sided *p*-value < 0.05 for all analyses, which were performed using Stata, version 13.1 (Stata Corp., College Station, TX, USA, 2013).

## 3. Results

### 3.1. Study Population Features

Overall, the number of registered visits was 11,733 over a 70-month period, of which 63.7% were made by male patients. The overwhelming majority were of Italian origin (82.3%), of a mean age of 34.5 yrs (standard deviation 12.8) (Table 1). Male gender and Italian origin were significantly more represented than female gender and non-Italian origin, respectively, in both pre-pandemic and pandemic periods.

The mean number of visits made each month was 163 overall. Before the pandemic, the mean number of visits s made each month was as high as 177, while after March 2020 the mean number decreased to 136 (−23.2%) (Figure 1). This trend is substantially conserved, even considering males and females separately.

Most of the patients who turned to the STI clinics were between the ages of 20 and 39 yrs; in more detail, 42.2% (4950) of the study patients were between 20 and 29 yrs and 23.9% (2800) between 20 and 39 yrs. The remaining age brackets were as follows: <20 yrs, 3.1%; 40–49 yrs, 17.8%; 50–59 yrs, 8.4%; and >60 yrs, 4.6%.

### 3.2. Visits by Season

When considering the whole period in the study, no significant difference in terms of rates of number of visits can be found between season; namely, they were 25.2% in spring, 25.0% in summer, 25.2% in autumn and 24.6% in winter (Table 1).

On the other hand, when taking into consideration the pre-pandemic and the pandemic time periods separately, the visits changed, even significantly, between seasons. More specifically, in the pre-pandemic period, visits to STI clinics increased in the autumn/winter months when compared to spring/summer, while the trend was the opposite in the pandemic period.

Comparing each season between the pre-pandemic and pandemic periods, a significant decrease was found in the latter period compared to before the pandemic (*p*-value < 0.001% for all seasons). The greatest decrease was found between pre-pandemic winters and those following the advent of the pandemic, namely February and March 2020 and winter 2021.

These findings are exactly superimposable, considering males (Table 2) and females separately (Table 3). The inter-seasonal variations found in the population as a whole, both in pre-pandemic and pandemic periods, were maintained in the two sexes separately. A remarkable finding is that the mean age of women after the advent of the pandemic was higher than in the pre-pandemic period.

### 3.3. Further Variables Potentially Conditioning Accesses

In terms of the variables selected, in the post-pandemic time period, the mean age of patients was significantly higher than in the pre-pandemic period (Table 1). There seems to be a significant increase in the visits made by older patients, especially 50+ years of age: 12.0% (4.0% for patients over 60 and 8.0% for patients between 50 and 59) in the pre-pandemic period, compared to 15.9% (6.6% for patients over 60 and 9.3% for patients between 50 and 59) in the pandemic period (Table 1). However, in general terms, the multivariate analysis did not reveal a significant difference in visits to STIs clinics based on the patients’ age (Table 4). The only variables conditioning the visits in a statistically significant way were pre-pandemic, compared to the pandemic period and seasonality. In particular, the decrease in visits after the advent of COVID-19 and the strong discrepancies among seasons were confirmed by the multivariate analysis.

## 4. Discussion

It is well known that a wide number of diseases do not occur randomly through the year, but exhibit preferred monthly or seasonal variations. For example, acute kidney injury in elderly patients, the occurrence of delirium in patients admitted to medical units, and many cardiovascular disorders, such as aortic disease, pulmonary embolism, venous thromboembolism, myocardial infarctions, transient ischemic attacks, heart failure and acute cardiovascular diseases [25,26,27,28,29,30,31,32,33]. Flare-ups of multiple sclerosis and peptic ulcer hospitalizations also show a seasonal pattern [34,35]. A seasonal variability has been found in extremely heterogeneous conditions, such as febrile seizures, acute pancreatitis, epistaxis, herpes zoster, acute microcrystalline arthritis, cranial nerve paralysis and migraine attacks in children [36,37,38,39,40,41,42].

Moreover, the COVID-19 pandemic impacted all aspects of healthcare, and its effects on many diseases were studied across the board by groups worldwide. For example, an Italian group found that the COVID-19 pandemic has changed the epidemiology of acute respiratory infections in children. In particular, respiratory syncytial virus (RSV) reached an extremely high peak during the pandemic, whereas the activity of the influenza virus was minimal. Before the pandemic, RSV and influenza had a clear seasonal pattern, but in 2020 this pattern was subverted [43].

A survey conducted at the Reims University Hospital, France, showed how the number of blood smears carried out for malaria dropped only in 2020, following the pandemic, and had no further variation in the following years; however, the seasonal (summer) increase in cases of malaria was preserved [44].

A Spanish group showed how the COVID-19 pandemic put the healthcare systems under such intense pressure that a decline in cerebrovascular disease (CDV) and acute myocardial infarction (AMI) daily hospital admissions was noted during the first two waves of the pandemic [45].

The COVID-19 pandemic has mainly affected the most vulnerable patients, such as patients with cancers or chronic diseases, as well as those who require ready access to healthcare services due to acute life-threatening conditions [46,47,48,49].

Moreover, the pandemic has led to late or missed diagnoses, because of delayed, or complete lack of, medical care [50,51,52].

To our knowledge, no study has ever focused on evaluating the seasonal trends in terms of patients’ visits to STIs clinics, especially in regard to pre- and post-pandemic differences.

A first noteworthy finding of this study was a predictable decrease in visits from the advent of the pandemic compared to before. The restrictions on socialization decreed in Italy in the months following the beginning of the pandemic are likely to have accounted for this finding. Therefore, in general, i.e., without stratifying by seasonality, attendance at outpatient clinics dedicated to the diagnosis and management of STIs has significantly reduced compared to the pre-pandemic period, probably due to a true reduction in the opportunities for sexual contagion. It should be remembered that the study clinics did not stop their activity, even during the months of lockdown. Thus, the decrease in the number of visits is not attributable to the reduction in, or suspension of, even temporarily, the activity of the STI clinics.

Our data show how the decrease in visits to STI clinics after the advent of the pandemic affected males and females almost equally (Table 4). Not only that, but the gap between genders, in terms of access to STI services, reduced with the pandemic (Table 1). In keeping with this, the graph lines shown in Figure 1 come closer in the pandemic period. This seems to be in disagreement with other studies, which reported a higher prevalence of healthcare disruption among women when compared with men [53,54]. Based on these data, in the study geographical area the pandemic did not lead to gendered discrepancies in terms of access to STI services.

With specific reference to seasonal variations, we found clear differences in the visits, both in the pre-pandemic and in the post-pandemic time period. However, interestingly, opposite trends were observed.

In detail, pre-pandemic, autumn and winter recorded increased visits to STI clinics compared to spring and summer. On the contrary, during the pandemic, autumn, and-above all, winter visits decreased very significantly in comparison both with pre-pandemic values of the same seasons and with spring and summer visits during the pandemic. This trend was clearly defined both considering the population as a whole and distinguishing between males and females.

The differences highlighted in the post-pandemic period for the autumn and winter season can be easily traced back to the more stringent self-isolation measures enforced during the colder months, coinciding with the spread of the COVID-19 infection. In this regard, it should be recalled that in Italy during the months of the peak of the COVID-19 contagiousness and spread, i.e., from November to March, public and meeting places were closed and citizens were discouraged (in 2021), if not prevented (in 2020), from dating. This certainly limited the possibilities of meetings, as well as new or casual sexual relationships, in the aforementioned periods.

On the other hand, before the pandemic, fluctuations in frequency of sexual activity had already been established by many studies through the analysis of various items, to some extent related to it, such as birth trends, medical abortion trends, HIV test trends, and more. They showed a bimodal trend with two peaks, namely one around Christmas and one in the summer time (The Summer Vacation Theory) [9,11,12,14,15]. The reported peak from Christmas to the New Year can be linked to the effect of festivals and gatherings that favor socialization and, in turn, sexual encounters, during this time of year. Increased alcohol consumption appears to be a further determinant, especially for unsafe sex [12].

The Summer Vacation Theory may be explained both by biological factors, namely seasonal hormonal changes that affect disposition and sexual interest, and sociocultural aspects, that is, the summer holidays [15].

The peak of sexual activity around Christmas, as described in the literature, seems to be in line with the increase in terms of numbers of visits observed in our study during the winter season, in the pre-pandemic period. The high number of visits recorded in autumn might be considered a consequence of the summer vacation.

With reference to the significant increase in the number of visits made by over 50 y/o patients during the pandemic, this could be due to the lesser restrictions that affected the workplace compared with other meeting places, such as bars, clubs and discos, which are more typically frequented by younger subjects. According to this interpretation, younger people may have suffered more from the restrictions imposed during the lockdown than the more elderly. It must be said that the available data on the access to primary and secondary STI prevention interventions during the pandemic by younger people are rather discordant, even within the same country [15,53,55,56]. Another possible explanation, at least partial, may lie in the unregulated environment of sex work, which is a prerogative of the older population as well.

In this study, a relative increase in patients of Italian nationality (81.1% vs. 85.5%) after the advent of the pandemic was also noted. The interpretation and significance of such finding is not straightforward, and still needs further investigation.

This study has several limitations. First, a diagnosis for each patient/visit was not addressed. Moreover, further variables, such as differentiation between nationalities of non-Italian patients and sexual orientation of each subject, were not included. These data could have given more insight and a better understanding in terms of stratification regarding both seasonality and pre- and post-pandemic times. The time span considered in the study did not allow for data on access to STI clinics in a post-pandemic phase. Continuation of monitoring could give indications on how the seasonality of visits varies with the reduction of the spread of COVID-19 and the collapse of the related social restriction measures, i.e., in a condition similar to the pre-pandemic period. An analysis of the health consequences of reducing access to STI outpatient clinics was beyond the objectives of the study, and was not assessed.

In conclusion, the present study included a considerable time span and a conspicuous number of cases, the analysis of which highlighted a clear difference in terms of number of visits to STI clinics based on the season; the overall reduction in numbers during the pandemic was also strongly highlighted.

Before the pandemic, the autumn and winter peaks in visits could probably be traced back to the well-recognized peaks of sexual activity in summer and during the Christmas period, respectively. On the other hand, the marked decrease during the pandemic, especially in the autumn and winter months, can be linked to the “lockdown”/self-isolation ordinances and social distancing measures. The advent of the pandemic has not led to differences in visits between genders, either in general terms or in relation to seasonal variations.

## Figures and Tables

**Figure 1 jpm-13-00731-f001:**
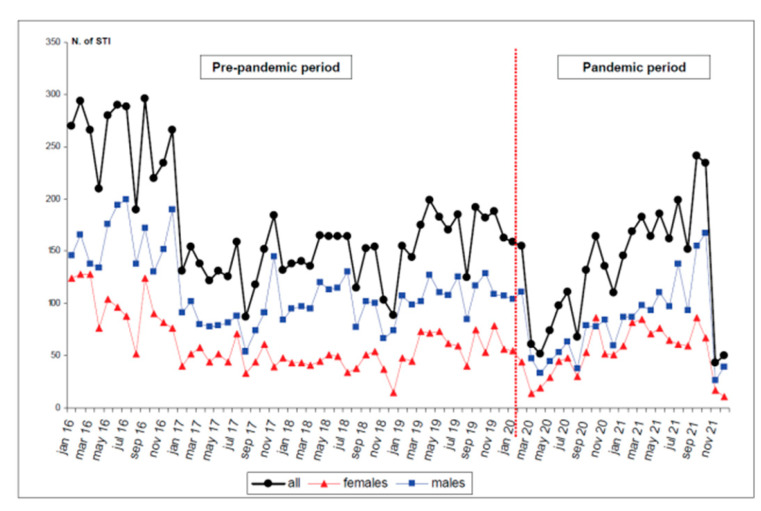
Trend in the total number of sexually-transmitted infections (STI) recorded between January 2016 and December 2021 in Bologna and Ferrara hospital wards, overall and by gender.

**Table 1 jpm-13-00731-t001:** Number of sexually-transmitted infections (STI) registered before (2016 to 2019) and during (2020 and 2021) the COVID-19 pandemic in Bologna and Ferrara hospital wards, overall and by selected epidemiological characteristics.

	Overall	Pre-Pandemic Period	Pandemic Period	*p* *
	*N* = 11,733	*N* = 8484	*N* = 3249	
Mean age in years (SD)	34.5 (12.8)	34.2 (12.4)	35.3 (13.7)	<0.001
Age class, % (*n*)				
- <20 yrs	3.1 (361)	3.1 (267)	2.9 (94)	0.6
- 20–29 yrs	42.2 (4950)	42.5 (3605)	41.4 (1345)	0.3
- 30–39 yrs	23.9 (2800)	23.7 (2009)	24.3 (791)	0.4
- 40–49 yrs	17.8 (2092)	18.7 (1588)	15.5 (504)	<0.01
- 50–59 yrs	8.4 (980)	8.0 (678)	9.3 (302)	0.02
- ≥60 yrs	4.6 (550)	4.0 (337)	6.6 (213)	<0.001
				<0.001
Male gender, % (*n*)	63.7 (7478)	64.8 (5494)	61.1 (1984)	
Female gender, % (*n*)	36.3 (4255)	35.2 (2990)	38.9 (1265)	
Nationality, % (*n*)				<0.001
- Italian	82.3 (9656)	81.1 (6877)	85.5 (2779)	
- Others	17.7 (2077)	18.9 (1607)	14.5 (470)	
Season, % (*n*)				
- Spring	25.2 (2954)	22.9 (1942)	31.1 (1012)	<0.001
- Summer	25.0 (2934)	22.2 (1886)	32.3 (1048)	<0.001
- Autumn	25.2 (2953)	26.0 (2205)	23.0 (748)	<0.001
- Winter	24.6 (2892)	28.9 (2451)	13.6 (441)	<0.001
		^b, c, d, e^	^b, c, d, e^	

SD: Standard deviation. Spring: 21 March–21 June; summer: 22 June–21 September; autumn: 22 September–21 December; winter: 22 December–20 March. * Chi-squared test and *t*-test for categorical and continuous variables, respectively. ^b^: *p* < 0.05 for the comparison between spring and autumn; ^c^: *p* < 0.05 for the comparison between spring and winter; ^d^: *p* < 0.05 for the comparison between summer and autumn; ^e^: *p* < 0.05 for the comparison between autumn and winter.

**Table 2 jpm-13-00731-t002:** Males only-Number of sexually-transmitted infections (STI) registered before (2016 to 2019) and during (2020 and 2021) the COVID-19 pandemic in Bologna and Ferrara hospital wards, overall and by selected epidemiological characteristics.

	Overall	Pre-Pandemic Period	Pandemic Period	*p* *
	*N* = 7478	*N* = 5494	*N* = 1984	
Mean age in years (SD)	36.1 (12.5)	36.1 (12.4)	36.3 (12.9)	0.5
Age class, % (*n*)				
- <20 yrs	2.3 (173)	2.5 (135)	1.9 (38)	0.13
- 20–29 yrs	35.7 (2673)	35.3 (1940)	37.0 (733)	0.2
- 30–39 yrs	25.0 (1872)	24.9 (1365)	25.6 (507)	0.5
- 40–49 yrs	22.4 (1671)	23.5 (1293)	19.1 (378)	0.001
- 50–59 yrs	9.6 (721)	9.3 (511)	10.6 (210)	0.09
- ≥60 yrs	4.9 (368)	4.6 (250)	6.0 (118)	0.014
Nationality, % (*n*)				0.7
- Italian	14.1 (1054)	14.2 (780)	13.8 (274)	
- Others	85.9 (6424)	85.8 (4714)	86.2 (1710)	
Season, % (*n*)				
- Spring	25.5 (1910)	23.4 (1283)	31.6 (627)	<0.001
- Summer	25.2 (1877)	22.6 (1241)	32.1 (636)	<0.001
- Autumn	25.2 (1886)	25.8 (1417)	23.6 (469)	0.053
- Winter	24.1 (1805)	28.3 (1553)	12.7 (252)	<0.001
		^b, c, d, e^	^b, c, d, e^	

SD: Standard deviation. Spring: 21 March–21 June; summer: 22 June–21 September; autumn: 22 September–21 December; winter: 22 December–20 March. * Chi-squared test and *t*-test for categorical and continuous variables, respectively. ^b^: *p* < 0.05 for the comparison between spring and autumn; ^c^: *p* < 0.05 for the comparison between spring and winter; ^d^: *p* < 0.05 for the comparison between summer and autumn; ^e^: *p* < 0.05 for the comparison between autumn and winter.

**Table 3 jpm-13-00731-t003:** Females only-Number of sexually-transmitted infections (STI) registered before (2016 to 2019) and during (2020 and 2021) the COVID-19 pandemic in Bologna and Ferrara hospital wards, overall and by selected epidemiological characteristics.

	Overall	Pre-Pandemic Period	Pandemic Period	*p* *
	*N* = 4255	*N* = 2990	*N* = 1265	
Mean age in years (SD)	31.6 (12.7)	30.8 (11.7)	33.6 (14.7)	<0.001
Age class, % (*n*)				
- <20 yrs	4.4 (188)	4.4 (132)	4.4 (56)	0.9
- 20–29 yrs	53.5 (2277)	55.7 (1665)	48.4 (612)	<0.001
- 30–39 yrs	21.8 (928)	21.5 (644)	22.5 (284)	0.5
- 40–49 yrs	9.9 (421)	9.9 (295)	10.0 (126)	0.9
- 50–59 yrs	6.1 (259)	5.6 (167)	7.3 (92)	0.03
- ≥60 yrs	4.3 (182)	2.9 (87)	7.4 (95)	<0.001
Nationality, % (*n*)				<0.001
- Italian	24.0 (1023)	27.7 (827)	15.5 (196)	
- Others	75.9 (3232)	72.3 (2163)	84.5 (1096)	
Season, % (*n*)				
- Spring	24.5 (1044)	22.0 (659)	30.4 (385)	<0.001
- Summer	24.8 (1057)	21.6 (645)	32.6 (412)	<0.001
- Autumn	25.1 (1067)	26.4 (788)	22.1 (279)	0.003
- Winter	25.6 (1087)	30.3 (898)	14.9 (189)	<0.001
		^b, c, d, e^	^b, c, d, e^	

SD: Standard deviation. Spring: 21 March–21 June; summer: 22 June–21 September; autumn: 22 September–21 December; winter: 22 December–20 March. * Chi-squared test and *t*-test for categorical and continuous variables, respectively. ^b^: *p* < 0.05 for the comparison between spring and autumn; ^c^: *p* < 0.05 for the comparison between spring and winter; ^d^: *p* < 0.05 for the comparison between summer and autumn; ^e^: *p* < 0.05 for the comparison between autumn and winter.

**Table 4 jpm-13-00731-t004:** Potential predictors of the number of visits due to sexually-transmitted infections (STIs) in the hospital wards of Bologna and Ferrara.

*Variables*	Regression Coeff. (95% CI)	*p*
Pandemic vs. pre-pandemic period *	−16.3 (−18.9; −13.8)	<0.001
Males vs. females	−2.19 (−4.62; 0.24)	0.08
Age-class:		
- <20 yrs	0 (ref. cat.)	--
- 20–29 yrs	4.12 (−2.57; 10.8)	0.2
- 30–39 yrs	3.12 (−3.76; 10.0)	0.4
- 40–49 yrs	3.91 (−3.14; 10.9)	0.3
- 50–59 yrs	4.66 (−2.93; 12.3)	0.2
- ≥60 yrs	−2.30 (−10.; 6.01)	0.6
Season:		
- Spring	0 (ref. cat.)	--
- Summer	−7.72 (−10.9; −4.52)	<0.001
- Autumn	−2.30 (−5.50; 0.90)	0.2
- Winter	7.04 (3.79; 10.3)	<0.001
Foreign vs. Italian nationality	−1.06 (−4.08; 1.94)	0.5

* Pre-pandemic period: January 2016–December 2019; pandemic period: January 2020–December 2021. Coeff.: coefficient; CI: confidence interval; ref. cat.: reference category; random-effect linear regression with one cluster unit (hospital ward).

## Data Availability

The data presented in this study are available on request from the corresponding author.

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
