# Peer review of "Visits to Sexually Transmitted Infection Clinics in Italy from January 2016 to November 2021: A Multicenter, Retrospective Study"

_jpm, 2023, doi:10.3390/jpm13050731_

Round 1

Reviewer 1 Report

The aim of this study is not obvious. Is there any suggestion or solution for dealing with this issue in the same pandemic? Did these results clarify current knowledge for coping with the same problems?

Is there any access to digital health or visual medicine for monitoring and caring of these patients at home in italy? If yes, is there any paper to study these topics? 

Reviewer 2 Report

Dear Authors, 

This is a well-written paper on the decrease in visits to STD clinics in Italy during the pandemic.  

I would consider changing the title and the focus from seasonality to purely Access to STI clinics in Italy...    I believe you rightly conclude that the decrease seen was primarily linked to increase in "lockdown" and social distancing when the spread of COVID was occurring and that if the seasonality of COVID had been different, your results would also be different.  For example the summer of 2020 also had a significant decrease in visits to the clinics in your study.  I believe the time period is too short to understand visit seasonality post-pandemic.

Round 2

Reviewer 2 Report

Dear Authors,

I believe your paper is improved and have no additional recommendations.

Best of Luck